# Three-Dimensional PLGA Nanofiber-Based Microchip for High-Efficiency Cancer Cell Capture

**DOI:** 10.3390/ma16083065

**Published:** 2023-04-13

**Authors:** Mengting Qi, Meilin Ruan, Jinjin Liang, Zhengtao Zhang, Chaohui Chen, Yiping Cao, Rongxiang He

**Affiliations:** Key Laboratory of Optoelectronic Chemical Materials and Devices of Ministry of Education, Institute of Interdisciplinary Research, Jianghan University, Wuhan 430056, China

**Keywords:** cancer cells, nanofibers, micropillar, mesoscopic interface, microchip

## Abstract

A 3D network capture substrate based on poly(lactic-co-glycolic acid) (PLGA) nanofibers was studied and successfully used for high-efficiency cancer cell capture. The arc-shaped glass micropillars were prepared by chemical wet etching and soft lithography. PLGA nanofibers were coupled with micropillars by electrospinning. Given the size effect of the microcolumn and PLGA nanofibers, a three-dimensional of micro-nanometer spatial network was prepared to form a network cell trapping substrate. After the modification of a specific anti-EpCAM antibody, MCF-7 cancer cells were captured successfully with a capture efficiency of 91%. Compared with the substrate composed of 2D nanofibers or nanoparticles, the developed 3D structure based on microcolumns and nanofibers had a greater contact probability between cells and the capture substrate, leading to a high capture efficiency. Cell capture based on this method can provide technical support for rare cells in peripheral blood detection, such as circulating tumor cells and circulating fetal nucleated red cells.

## 1. Introduction

Circulating tumor cells (CTCs), which are shed from the primary tumor and invade peripheral blood, play an important role in cancer metastasis [1]. CTCs can escape from the blood circulation system and establish distant metastasis in other organs, inducing cancer metastasis and recurrence; CTCs account for 90% of cancer-related deaths [2]. The genotype and phenotype information of CTCs are similar to those of primary tumors; therefore, CTCs are the ideal liquid biopsy targets for cancer early diagnosis, disease screening and therapy guidance. The numbers of CTCs captured from the patient’s peripheral blood can be used as a reference value to assess the patient’s condition, such as the risk of cancer and recurrence [3]. Compared with the prostate specific antigen, CTC number was also used as a response measure of prolonged survival for metastatic prostate patients [4]. In addition, the relationship and interaction between CTCs and immune cells or neutrophils is important for the development of new cancer therapies and may be the critical factor why cancer is difficult to permanently cure [5]. Therefore, capturing the CTCs from the patient’s peripheral blood is necessary.

Compared with the healthy blood cells, CTCs are rare, with only a few cells found in 1 mL of blood. In recent decades, different technologies have been developed to capture CTCs [6], including antibody label-free substrates [7,8], antibody-labeled substrates, and antibody-like labeled substrates, such as polypeptides [9], aptamers [10], and DNA [11,12]. For cancer cell capture, different substrates have been fabricated. They can be divided into three classes. First, nanomaterials, such as nanofibers, nanoparticles, or nanosheets, were fabricated on the substrates to intensify the surface roughness to increase the contact probability between cells and substrates, and increase the capture efficiency [13,14]. Poly(lactic-co-glycolic acid) (PLGA) nanofibers or titanium dioxide nanofibers were electro-spun on substrates to form a network to capture cells [15,16,17]. The thickness of nanofibers can be controlled by the electrospinning time. The longer the electrospinning time, the thicker the nanofiber film. Titanium dioxide or manganese dioxide nanoparticles were fabricated on substrates to form a roughness cell capture substrate [18,19]. Graphene oxide nanosheets were used to specific capture CTCs, showing great sensitivity and a low background of white blood cells [14,20]. Natural cancer cell membranes were used as bio-interface to capture CTCs, demonstrating good antiadhesion properties for white blood cells due to their natural electronegativity [21]. However, the films were still two-dimensional and the heights remained limited. The thickness of the nanosheet may be less than 2 nm, and the diameter of nanoparticle may be less than 500 nm. Therefore, compared to the size of cancer cells, these substrates may be flat.

Second, the size characterization of cells, micropillars [22], micropores [23], and microstructure traps [24] were used to capture cancer cells. The shapes of micropillars varied and the cell capture efficiency was also different. Micropillars of equilateral triangle shape integrated into a microchip were used as a size-dictated immunocapture chip, which utilized deterministic lateral displacement [25]. Compared to the circular micropillars, these triangle-shape micropillars can provide a gradient of hydrodynamic forces, which can address the problem of different expression levels of surface antigen of individual cancer cells. Triangle-shaped micropillars were also designed with unique geometries to specifically differentiate CTC clusters [24]. Advanced nanomaterials were integrated with microparticles, micropillars, or other microstructures to form a hierarchical bio-interface that simulates the characterization of cancer cells’ micro- and nano-membrane to increase the cancer cell capture efficiency [26]. Gold nanoparticles were coated on the surface of microfluidic chip to enhance capture efficiency and recovery of isolated CTCs [27]. In order to enlarge the size difference between CTCs and health white blood cells, CTCs were captured on specific biomarker modified microparticles, which can be efficiently purified and released [28]. However, these cell capture substrates usually need an additional microstructure to enhance the contact probability between cells and substrate surfaces. Therefore, it is necessary to develop a three-dimensional microfluidic chip for cancer cell capture. Utilizing a quadratic-sacrificing template method, a three-dimensional poly(dimethylsiloxane) (PDMS) scaffold was embedded into the microchannel to enable fast isolation [29]. It was noted that the PDMS scaffold was spatially distributed in the microchannel, which could generate fluid chaotic flow so that cancer cells had more binding sites. Further, thermosensitive gelatin was modified on the three-dimensional PDMS scaffold by layer-by-layer assembly, which can be used as a functional material to capture and release cancer cells [30]. The three-dimensional PDMS scaffold was further fabricated as an injectable and retractable probe, which can be injected into patient blood vessel to intravascularly capture cancer cells in vivo [31]. These previously reported studies indicated that spatially 3D microstructures can increase the cancer cell capture efficiency due to the increased contact binding sites between cells and substrates.

In this work, a micropillar–nanofiber hierarchical 3D topological substructure was designed in microfluidic chips and used for cancer cell capture. In order to distinguish from the cylindrical micropillar, arc-shaped glass micropillars were designed and fabricated through chemical wet etching and conventional soft lithography. Micropillars fabricated using conventional soft lithography and transformation were cylindrical [32]. Compared to the cylindrical micropillars, arc-shaped micropillars had increased contact binding sites. Therefore, in this work, chemical wet etching was utilized to fabricate lateral side surfaces of micropillars. PLGA nanofibers were electrospun on the arc-shaped micropillars to from a 3D network in the microchannel as shown in Figure 1. When the cancer cells are injected and flow in the microchannel, the contact probability increases and the specific antibodies grafted on nanofibers can capture the cells. Therefore, this micro-nano hierarchical cell capture substrate assembled by a nanofiber network and micropillars in the microchannel can benefit the specific recognition of cancer cells in a dynamic cell recognition process. We believe that this cell capture technology may provide selective guidance to develop new hierarchical bio-interfaces for rare cell detection in the future.

## 2. Materials and Methods

### 2.1. Materials

Ethanol, acetone, hydrofluoric acid, nitric acid, ammonium fluoride, sodium hydroxide, glutaraldehyde (25%) and dimethylformamide (DMF) were purchased from Sinopharm Chemical Reagent Co., Ltd., Shanghai, China. Ceric ammonium nitrate, perchloric acid, tetrahydrofuran (THF), and 2-(N-morpholino)-ethanesulfonic acid were acquired from Shanghai Aladdin Biochemical Technology Co., Ltd., Shanghai, China. Polydimethylsiloxane (PDMS, RTV615) was obtained from Momentive, New York, NY, USA. AZ5214 and SU8 3050 photoresists were purchased from Microchem, Round Rock, TX, USA. DI water was obtained from Milli-Q system (Millipore, Burlington, MA, USA). DMEM, phosphate-buffered saline (PBS), and streptomycin penicillin were purchased from Thermo fisher Scientific Inc, Waltham, MA, USA. Streptavidin and fetal bovine serum were acquired from Invitrogen, Waltham, MA, USA. IFkine green donkey anti-goat IgG was purchased from Abbkine, Wuhan, China. Biotinylated anti-human epithelial cell adhesion molecule (anti-EpCAM) antibody was purchased from R&D systems, Minneapolis, MN, USA. Fluorescein diacetate (FDA), 1-(3-dimethylamino-propyl)-3-ethylcaarbon diimmide (EDC), and N-hydroxysuccinimide (NHS) were purchased from Sigma-Aldrich, St. Louis, MI, USA. Poly(lactic-co-glycolic acid) (PLGA) was acquired from Jinan Daigang Biomaterial Co., Ltd., Jinan, China. All reagents were used without additional purification.

### 2.2. Microchip Design and Fabrication

The cancer cell capturing microchips contained a microfluidic channel and a spatial 3D substrate coupled with micropillars and PLGA nanofibers. As shown in Figure 1a, the microchannel was fabricated by standard soft lithography according to previously reported work [22,33]. First, the design of the microchannel was transferred to a photo mask using a UV laser direct photolithography system (μPG 501, Heidelberg Instruments, Heidelberg, Germany). Second, the silicon wafer was baked on a hot plate at 105 °C for 10 min to remove the surface water. Then, SU8 3050 photoresist was spin-coated onto the silicon wafer at a speed of 2800 rpm for 30 s and then baked on a hot plate at 95 °C for 45 min. After UV exposure, the photoresist was post-baked at 65 °C for 1 min and 95 °C for 4 min, respectively. Then, the photoresist was developed by SU8 developer solution and the mode of the microchannel was fabricated. The height and width of the microchannel were 50 μm and 5 mm, respectively. PDMS prepolymer (base-polymer: cross-linker = 10:1) was poured onto the microchannel mold, and then cured on a hot plate at 85 °C for 2 h after being degassed using a negative pressure pump. After being peeled from the mold, the PDMS microchannel was punched with inlet and outlet holes.

As shown in Figure 1b, the micropillar array was fabricated through photolithography and wet etching. The micropillar array design was also transferred to the glass-based photomask by utilizing μPG 501 (a chromium-based substrate, SG3006, Changsha Shaoguang Tec., Changsha, China). The dimension of the circle was 50 μm. The substate was then wet-etched for 15 min in a solution (2 mol L^−1^ HF, 2 mol L^−1^ NH_4_F and 2 mol L^−1^ HNO_3_, volume ratio was 2:1:1). The etched substrates were washed by acetone, chromium removal solution, and ethanol to remove the remaining photoresist and chromium. PLGA was dissolved by magnetic stirring in a DMF and THF mixing solution (DMF: THF 3:1, *v*/*v*) with a concentration of 15% (*w*/*v*). PLGA nanofibers were electrostatically spin-coated on the etched substrate to couple the micropillar with PLGA nanofibers to form a spatial 3D substrate. The high voltage was set at 15 kV and the distance between positive and negative electrodes was about 13 cm. The flow volume of PLGA solution was set at about 100 μL h^−1^, controlled by an injection pump. The density of the nanofiber was controlled by the electro-spinning time, which was set at 2 min, 4 min, and 6 min, respectively. Finally, the PDMS microchannel was bound to the PLGA nanofiber coupled with the wet-etched arc-shaped glass micropillar substrate using a plasma treatment instrument, as shown in Figure 1c. The fabricated microchips were baked at 60 °C for 24 h prior to anti-EpCAM antibody modification to increase the bonding strength.

### 2.3. Specific Antibody Modification in Microchannels

In order to capture the cancer cells, a specific anti-EpCAM antibody was chemically modified on the microchip [15]. As shown in Figure 1d, EDC and NHS in MES buffer solution (0.4 M EDC and 0.6 M NHS in 0.3 M MES) were injected into the microchannel using a Lange syringe pump at a flow velocity of 300 μL h^−1^ for 1 h at room temperature to activate the carboxyl group. After washing with PBS at a flow velocity of 300 μL^−1^ for 15 min, SA solution (50 μg mL^−1^ in PBS solution) was injected into the microchannel at a flow velocity of 500 μL h^−1^ for 2 h. After washing with PBS, biotinylated anti-EpCAM antibody solution (10 μg mL^−1^ in PBS solution) was injected into the microchannel at a flow velocity of 50 μL h^−1^ for 2 h at room temperature. Finally, the microchannel was washed using PBS solution.

### 2.4. Cancer Cell Capture and Detection

EpCAM positive cancer cells MCF-7 were used to investigate the cell capture efficiency using this microchip comprised of 3D micropillar coupled with PLGA nanofibers. The cells were cultured in DMEM with 3% FBS and 1% penicillin-streptomycin in a cell incubator (37 °C, 5% CO_2_). MCF-7 cells were diluted to different concentrations in DMEM culture medium for cell capture study. The cancer cells were fluorescence-stained by FDA to identify the cancer cells captured by the anti-EpCAM antibody in the microchannel. The number of cells captured in the microchannel was counted using an inverted fluorescence microscope (Nikon Eclipse Ti-s, Tokyo, Japan). The cell capture efficiency can be calculated as follows [22]:$$I_{c}=\frac{N_{c}}{N_{c}+N_{W}}$$
where $N_{c}$ is the number of cancer cells captured in the microchannel, and $N_{w}$ is the number of cancer cells that were not captured. 

### 2.5. Scanning Electron Microscopy (SEM) Characterization

The morphologies of chemical etched micropillars and electrospinning PLGA nanofibers were characterized using SEM (Hitachi, S3400, Tokyo, Japan). For the morphology characterization of the captured cells, cells were incubated with 4% glutaraldehyde overnight to fix the cell skeleton and then gradient dehydrated sequentially in ethyl alcohol with concentrations of 30%, 50%, 70%, 80%, 90%, 95% and absolute ethyl alcohol [34]. Then, the cells were freeze-dried for 1h. After gold sputtering, SEM was used to characterize the capture cells morphologies on the three-dimensional substrates.

## 3. Results and Discussion

### 3.1. Microfluidic Chips

The structure of microchips is important for cancer cell capture and can influence the fluid and cell flow states [35,36,37]. For static cell capture, cancer cells should be sedimented down to the substrate so they can be in contact with the antibody. Therefore, the cell capture efficiency is affected by the cell sediment time, which takes about 45 min to reach the highest capture efficiency [16,18]. Cell capture substrates integrated with microchannel and nanoparticles or nanopillars need a mixer on the top of microchannel to increase the contact probability of cells and antibodies [22]. In this work, a 3D spider web biomimetic cell capture PLGA nanofibers coupled with micropillars were integrated into the microchannel, as shown in Figure 2a. The micropillar arrays and electrospun PLGA nanofibers are illustrated in Figure 2b. 

Chemical wet etching was used to fabricate the micropillars, as shown in Figure 3a,b. Compared with those prepared by the soft-lithograph method, the lateral side of the micropillars fabricated by wet etching was arc-shaped due to the isotropous etching of the glass. SEM images showed that the top surface dimension, bottom dimension, and height of the micropillar are about 20, 50 and 30 μm, respectively. Meanwhile, cancer cells are spheroidal, and the sizes of MCF-7 cells are about 15 μm. Therefore, compared with the cylinder-shaped micropillar, the contact probability of cancer cells to micropillars increased due to the enlarged lateral side surface area. 

Cancer cells have many nanoscale structure features, such as microvilli and filopodia. Owing to the dimension coupling effect between cells and substrates, PLGA nanofibers were coupled on the etched glass micropillars, as shown in Figure 3c,d. The PLGA nanofibers can form 3D network structures around the arc-shaped glass micropillars. The size of the PLGA nanofibers was about 350 nm. In the flat area, PLGA nanofibers were on the glass surface and the height of nanofibers may be less than 2 μm, which was controlled by the electrospinning time. Utilizing the micropillars, it was shown that the PLGA nanofibers were supported and the distance from the substrate may be increased to 30 μm, which was controlled by the height of micropillars. In this method, cancer cells can be captured by the spatial distributed PLGA nanofibers before settling to the glass surface.

### 3.2. Specific Antibody Modification and Cancer Cell Capture

Owing to the size coupling effect, cancer cells can be non-specifically adsorbed on the nanosized substrate. However, this process usually takes a long time and the cell capture efficiency is low. In this work, a specific antibody, anti-EpCAM antibody, was chemically modified on the PLGA nanofiber surface, so that the PLGA nanofiber network in microchannel can capture cancer cells with high efficiency. IFKine green affinipure donkey anti-goat IgG was used to ensure that the antibody was successfully modified on the PLGA nanofibers. As shown in Figure 4a, no green fluorescence appeared when the anti-EpCAM antibody was not modified. On the contrary, green fluorescence was found on the substrate, indicating that the antibodies were successfully chemically modified on the substrate, as shown in Figure 4b. Therefore, EpCAM positive expression cancer cells can be captured on this PLGA nanofiber-coupled micropillar substrate, as shown in Figure 4c,d, which are the bright field and fluorescence image of the captured cancer cells, respectively.

After the cells were captured on the substrates, SEM was used to characterize the cell morphology. The cells were immobilized by glutaraldehyde for 15 min and then dehydrated sequentially with ethanol at different concentrations, from 30% to 95%, and absolute alcohol. After freeze-drying for 1 h, the cell morphology was characterized using SEM. Figure 5 shows the cancer cells coupled with the PLGA nanofibers, and the microvilli and filopodia of cells attached to the glass surface and PLGA nanofibers, which are labeled by red arrows in Figure 5d. From the side view of captured cells in Figure 5b,d, this 3D substrate can provide a suitable microenvironment for cancer cells. 

### 3.3. Effect of PLGA Nanofibers and Substrates on Cell Capture Efficiency

The density of PLGA nanofibers can be controlled by adjusting the electrospinning time. As shown in Figure 6a, the cancer cell capture efficiency was about 51% in the absence of PLGA nanofibers ($N_{c}$ and $N_{w}$ are 111 and 108, 198 and 210, 147 and 130, respectively). When the PLGA nanofibers electrospinning time was 2 min, the capture efficiency was increased to 54% ($N_{c}$ and $N_{w}$ are 197 and 169, 370 and 271, 284 and 275, respectively). When the electrospinning time was 4 min, the capture efficiency was increased to 91% ($N_{c}$ and $N_{w}$ are 1081 and 100, 876 and 57, 998 and 143, respectively). However, when the electrospinning time was increased to 6 min, the capture efficiency was not significantly increased. Therefore, prolonging the electrospinning time leads to an increase in the density of PLGA nanofiber and, consequently, the contact probability between the cells and nanofibers. However, when the electrospinning time continues to increase, the density of PLGA nanofibers is excessively large so that the contact probability decreases, leading to a reduction in cell capture efficiency.

The topological microstructure of the cell capture substrates is important for the detection of rare cells. As shown in Figure 6b, different kinds of substrates were used to investigate the effect of topological microstructures on the cell capture efficiency. For the flat glass substrate coupled with 4 min electrospun PLGA nanofibers, about 60% of EpCAM-positive cancer cells were captured. For the wet-etched micropillars without PLGA nanofibers, only 51% of cancer cells were captured on the substrate. After antibody modification, no carboxy group was found on the glass surface. The reason for the cells captured on the glass substrate may be due to the nanoscale surface after chemical wet etching, which can induce cancer cell capture [7]. When the wet-etched micropillar substrates were coupled with PLGA nanofibers, the efficiency could be increased to 91%. These results indicated that the 3D PLGA nanofiber network was the most effective substrate for cancer cell capture. According to previous work, cancer cells are sensitive to the substrate microenvironment and deformability [37]. Figure 6c shows that when the substrate was changed from the flat substrate coupled with PLGA nanofibers to wet-etched micropillars to wet-etched micropillars coupled with PLGA nanofibers, and the contact site was increased.

## 4. Conclusions

In this work, we developed a 3D microchip for dynamic cancer cell capture. Compared with traditional micropillars, chemical isotropic wet-etched glass micropillars have an arc-shaped lateral surface that can enlarge the contact site. Moreover, PLGA nanofibers were coupled to the wet-etched glass micropillars through electrospinning. Through this method, PLGA nanofibers can form a 3D structure in the microchannel so that cancer cells can be captured effectively at a rate of higher than 90%. Therefore, this substrate is suitable for the detection of rare cells, such as CTCs and fetal nucleated red blood cells in the peripheral blood. 

## Figures and Tables

**Figure 1 materials-16-03065-f001:**
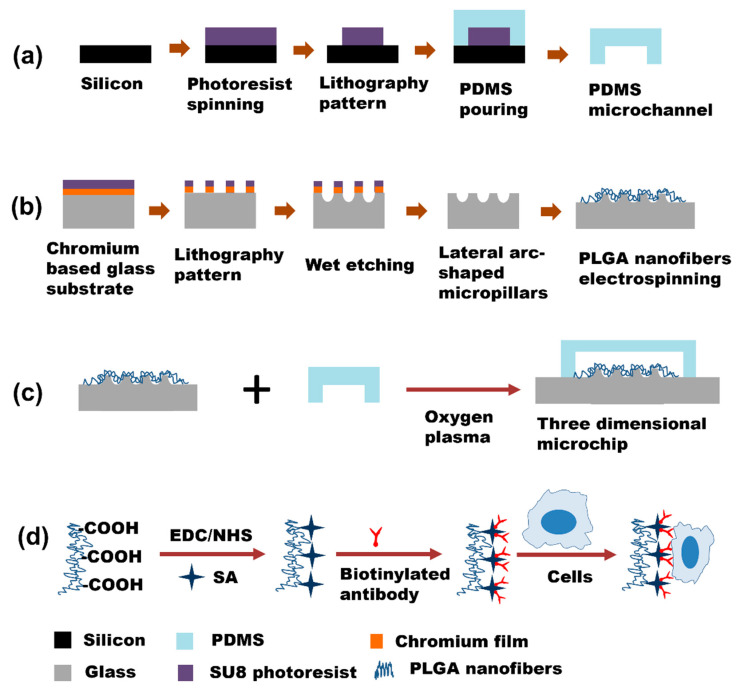
(**a**) PDMS microchannel fabrication. (**b**) Wet etching of lateral arc-shaped micropillars coupled with PLGA nanofibers. (**c**) PDMS microchannel was bound to substrate by oxygen plasma. (**d**) Specific antibody chemical modification on PLGA nanofibers.

**Figure 2 materials-16-03065-f002:**
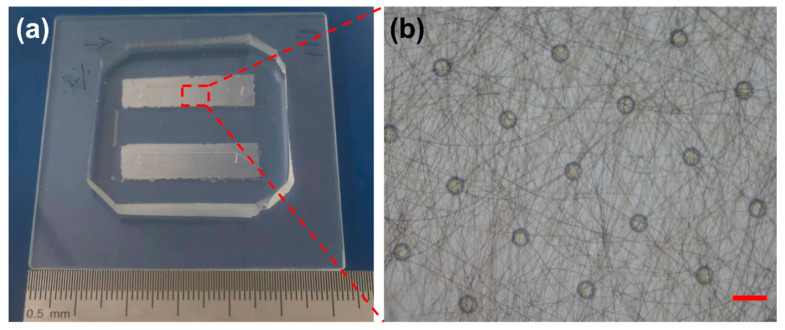
Cancer cell capture microchip (**a**) and PLGA nanofibers coupled with micropillars in the microchannel (**b**). The scale bar is 100 μm.

**Figure 3 materials-16-03065-f003:**
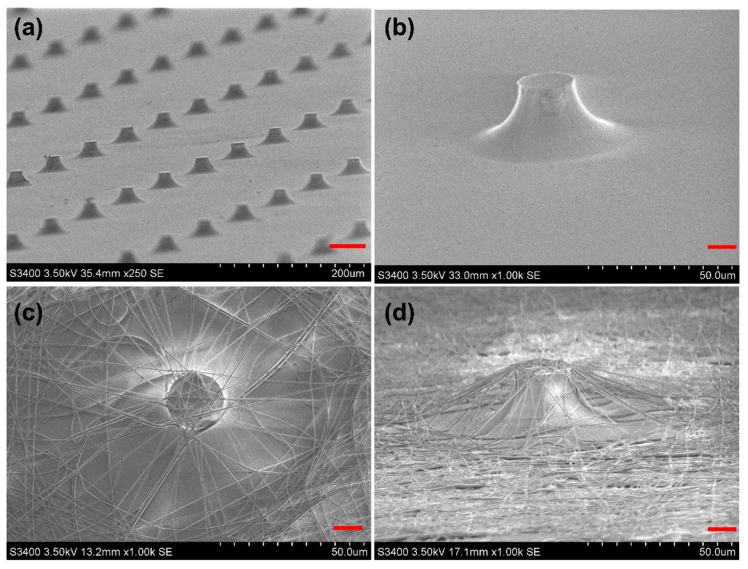
SEM images of the chemical wet-etched micropillars and the PLGA nanofibers. (**a**,**b**) Chemically wet-etched micropillars. The scale bar in (**a**) is 50 μm. (**c**,**d**) Electrospun PLGA nanofibers. The scale bars in (**b**–**d**) are 10 μm.

**Figure 4 materials-16-03065-f004:**
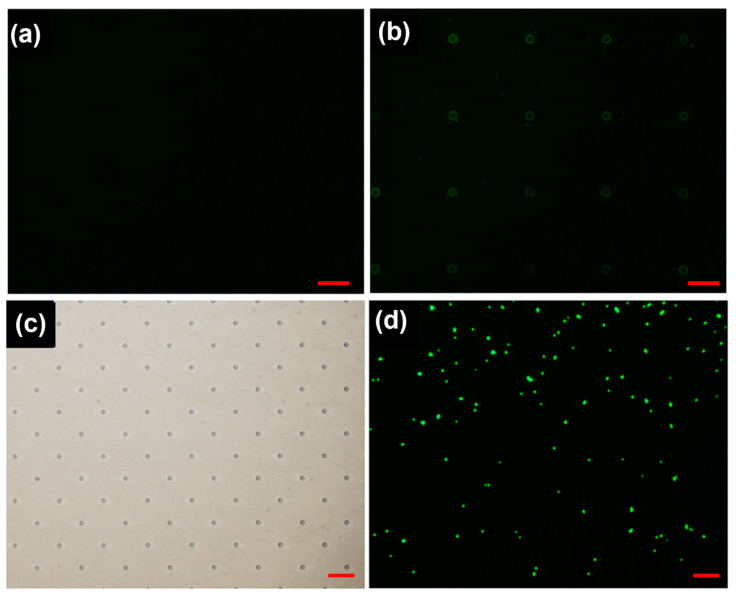
(**a**,**b**) Verification of a specific antibody modified on the substrates. (**c**) Bright field and (**d**) fluorescence field of the captured cells. The scale bars are 100 μm in (**a**,**b**) and 200 μm in (**c**,**d**).

**Figure 5 materials-16-03065-f005:**
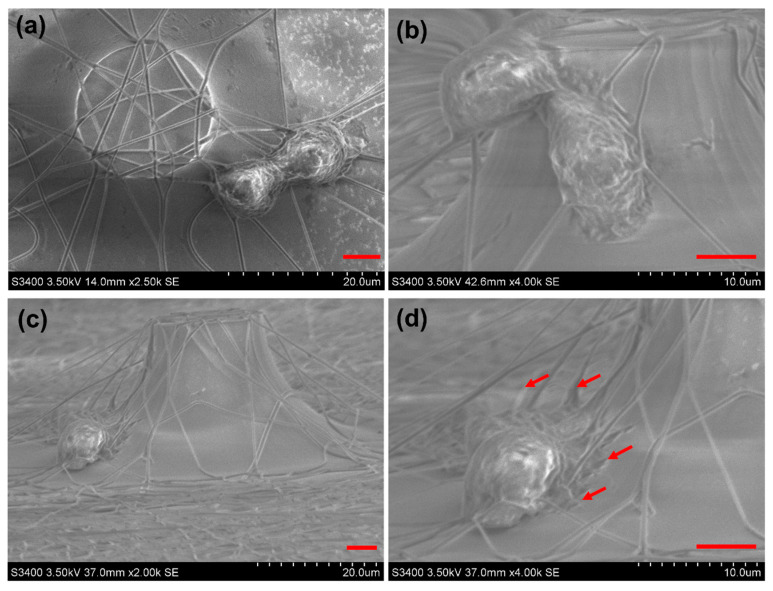
SEM characterization of captured cancer cells on the micropillars coupled with the PLGA nanofibers substrate. The top view (**a**) and the lateral view (**c**) of captured cells. (**b**,**d**) are the enlarged view of captured cells. The red arrows in (**d**) indicate the microvilli and filopodia of cells. The scale bars are all 5 μm.

**Figure 6 materials-16-03065-f006:**
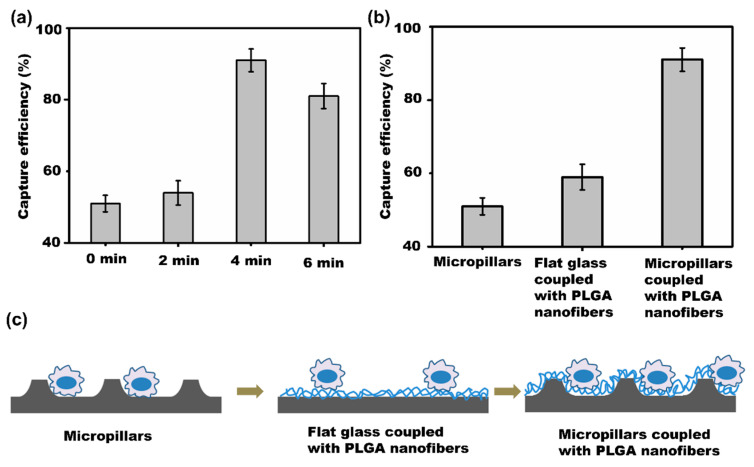
Effect of PLGA nanofibers electrospinning time (**a**) and different substrates (**b**) on the cancer cell capture efficiency. (**c**) Scheme of the contact site of cancer cells on the different substrates.

## Data Availability

Not applicable.

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
