# Peer review of "Three-Dimensional PLGA Nanofiber-Based Microchip for High-Efficiency Cancer Cell Capture"

_materials, 2023, doi:10.3390/ma16083065_

Round 1
Reviewer 1 Report
This is an interesting article where the authors have focused on the development of a 3D network capture substrate based on PLGA nanowires is studied. The glass micrometre column was prepared by soft lithography. PLGA nanofibers were coupled with micropillars by electrospinning. The following corrections are needed:
-In the introduction clearly mention how the work is different from the existing literature
-Specific antibody modification and cancer cells capture: This section needs more discussion with reference to the recent work
-SEM should reflect the actual magnification images. Its not showing currently
-Make sure error-bar graphs are there in the figures (e.g. 6)
Author Response
-In the introduction clearly mention how the work is different from the existing literature.
Response: Thanks very much for the reviewer's suggestion. In this work, a kind of three-dimensional substrate was fabricated, which integrated PLGA nanofibers and arc-shaped glass micropillars. The PLGA nanofibers on the top of micropillars can be hold up, which formed a space network in the microchannel. The difference can also be seen in the figure 6c. In the introduction, we had reviewed the exist technologies, which can support our standpoint.
-Specific antibody modification and cancer cells capture: This section needs more discussion with reference to the recent work
Response: Thanks very much for the reviewer's suggestion. We had added some discussion in the section.
-SEM should reflect the actual magnification images. It’s not showing currently
Response: Thanks very much for the reviewer's suggestion. We had shown the detail information about the SEM images in the revised manuscript.
-Make sure error-bar graphs are there in the figures (e.g. 6)
Response: Thanks very much for the reviewer's suggestion. The error bars had been shown in Fig.6.
Reviewer 2 Report
The manuscript entitled “Three dimensional PLGA nanofibers-based microchip for high efficiency cancer cells capture” by Qi Mengting , Ruan Meilin , Liang Jinjin , Zhang Zhengtao , Chen Chaohui , Cao Yiping , He Rongxiang is focused on PLGA nanowires application for cancer cells capturing. The goal is certainly relevant and the research paper can be published after revision. Please, find my comments below.
Please, check with journal guide for authors how should references be inserted into the text (square brackets or superscript, but definitely not the number next to the word)
Abbreviation PLGA needs to be deciphered in the introduction section.
I believe that the paragraph lines 86-100 looks like a conclusion.
What is the accuracy of capture efficiency determination?
«When the electrospinning time increased, the density of PLGA nanofibers increased» how was it controlled?
Author Response
Please, check with journal guide for authors how should references be inserted into the text (square brackets or superscript, but definitely not the number next to the word)
Response: Thanks very much for the reviewer's suggestion. In the process of writing manuscript, the number was inserted superscript in the text. The reason that references numbers were appeared next to the word may be due to something wrong in the arrangement process. According to the suggestion, we checked all the references in the revised manuscript.
Abbreviation PLGA needs to be deciphered in the introduction section.
Response: Thanks very much for the reviewer's suggestion. We had changed the abbreviation PLGA by complete spelling.
I believe that the paragraph lines 86-100 looks like a conclusion.
Response: Thanks very much for the reviewer's suggestion. According to the suggestion, we had revised this section, which may be suitable for introduction.
What is the accuracy of capture efficiency determination?
Response: Thanks very much for the reviewer's suggestion. As shown in the experiments and methods section 2.4, the cells capture efficiency was calculated through the number of cells captured in the microfluidic channel and the cells flow out of the microchannel. The cells were fluorescence stained by FDA. During the experiment, fluorescence microscopy was used to count the un-captured cells, which flows out of the microchannel. When the capture experiment finished, numbers of cells captured in the microchannel was counted using the fluorescence microscopy. Therefore, the cell capture efficiency can be accuracy calculated using the equation as shown in the manuscript.
When the electrospinning time increased, the density of PLGA nanofibers increased how was it controlled?
Response: Thanks very much for the reviewer's suggestion. PLGA nanofibers were fabricated using a high-voltage controlled electrospinning system. The volume of PLGA was set at 0.1 ml h-1 and the voltage was set at 15 kV. The distance between positive and negative electrodes was about 13 cm. The electrospinning time was controlled by turn on or turn off the high-voltage source. The electrospinning time was equaled to the applied high voltage time. Therefore, the density of PLGA nanofibers increased when the electrospinning time increased.
Reviewer 3 Report
The article titled “Three dimensional PLGA nanofibers-based microchip for high efficiency cancer cells capture” by Qi et al. is an interesting work that describes the use of a novel PLGA-nanowire micropillar assembly for capturing MCF-7 cancer cells. There are still some questions that need clarification and suggestions to strengthen the manuscript further:
1. There seems to be some inconsistency in the title with respect to the content seeing that nanofibers are very different compared to the nanowires. While the authors called the PLGA network as nanofibers in the Title of the manuscript, they refer to the same network as nanowires in the Introduction. It is suggested that authors correct for this inconsistency.
2. First paragraph in the Introduction section doesn’t make much sense. The authors need to elaborate on it and check for grammatical and spelling errors. I am assuming that the numbers in the paragraphs are the citations, which if so, should be modified for the correct referencing style and indentations.
3. In the abstract on line 10, the authors claim that they used soft lithography for the fabrication of the device, whereas towards the end in the Introduction section, lines 88-89, they claim that they used chemical wet etching instead of conventional soft lithography. This apparent conflict need to be resolved.
4. Figures 2 to 5 give a good evidence of how the device was fabricated and modified/ tested for the cancer cell capture.
5. To verify the use of cell capture efficiency equation on line 176, it is suggested that the authors provide raw data on the number of cells that were counted and how many times was this procedure repeated. Based on this information, it is suggested that the authors do some statistical analysis on the raw data and conclusively show the evidence of the resulting % efficiencies mentioned in the bar graphs 6a and 6b on page 9. Based on the statistical analysis, it is also suggested that the authors show error bars on the bar graphs.
6. The number of cells cultured versus the number of cells used in the microchannel is also missing, which the authors should provide the data for.
7. The dispersion and electrospinning protocol for PLGA is missing in the Materials Section. It is suggested that the authors add this piece of information to provide additional information on the synthesis of the PLGA nanofibrous mesh over the micropillars.
Author Response
- There seems to be some inconsistency in the title with respect to the content seeing that nanofibers are very different compared to the nanowires. While the authors called the PLGA network as nanofibers in the Title of the manuscript, they refer to the same network as nanowires in the Introduction. It is suggested that authors correct for this inconsistency.
Response: Thanks very much for the reviewer's suggestion. Exactly, nanofibers were different from nanowires. According to the suggestion, we had changed nanowires by nanofibers in the revised manuscript.
- First paragraph in the Introduction section doesn’t make much sense. The authors need to elaborate on it and check for grammatical and spelling errors. I am assuming that the numbers in the paragraphs are the citations, which if so, should be modified for the correct referencing style and indentations.
Response: Thanks very much for the reviewer's suggestion. In the first paragraph, we introduced some information about circulating tumor cells, which is shed from the primary tumor and invades into the peripheral blood system. Due to the rare characterization, it is necessary to capture and detect CTCs using different methods. The numbers in the manuscript are the citations. There may be some thing wrong during the translation from word to PDF. According to the reviewers’ suggestion, we had checked the referencing style in the revised manuscript.
- In the abstract on line 10, the authors claim that they used soft lithography for the fabrication of the device, whereas towards the end in the Introduction section, lines 88-89, they claim that they used chemical wet etching instead of conventional soft lithography. This apparent conflict needs to be resolved.
Response: Thanks very much for the reviewer's suggestion. Different expressions can lead to misunderstandings. In this work, the PDMS microchannel was fabricated using conventional soft lithography. As for the micropillars, cylindric micropillars can be fabricated using conventional soft lithography, such as PDMS micropillars. In this work, arc-shaped glass micropillars were fabricated, which can be fabricated easily using chemical wet etching method. The pattern of glass micropillars were controlled using conventional soft lithography. According to the reviewer’s suggestion, we described more detail here in the revised manuscript.
- Figures 2 to 5 give a good evidence of how the device was fabricated and modified/ tested for the cancer cell capture.
Response: Thanks very much for the reviewer's recognition.
- To verify the use of cell capture efficiency equation on line 176, it is suggested that the authors provide raw data on the number of cells that were counted and how many times was this procedure repeated. Based on this information, it is suggested that the authors do some statistical analysis on the raw data and conclusively show the evidence of the resulting % efficiencies mentioned in the bar graphs 6a and 6b on page 9. Based on the statistical analysis, it is also suggested that the authors show error bars on the bar graphs.
Response: Thanks very much for the reviewer's suggestion. According to the reviewer’s suggestion, we had given the error bars in figure 6a and 6b. The number of cells were in the range of 200 to 1200. As for the substrate in the absence of PLGA nanofibers, and are 111 and 108, 198 and 210, 147 and 130, respectively. For the substrates with 2 minutes electrospinning PLGA nanofibers, and are 197 and 169, 370 and 271, 284 and 275, respectively. For the substrates with 4 minutes electrospinning PLGA nanofibers, and are 1081 and 100, 876 and 57, 998 and 143, respectively. For the substrates with 6 minutes electrospinning PLGA nanofibers, and are 228 and 55, 490 and 102, 381 and 121, respectively.
- The number of cells cultured versus the number of cells used in the microchannel is also missing, which the authors should provide the data for.
Response: Thanks very much for the reviewer's suggestion. Usually, the cell density for cells culture on different substrate was about 1.0 e5 mL-1. Due to the inaccuracy of the cells number and cells sedimentation during the experiment, we just count the cells captured in the microfluidic channel and the cells flow out of the channel. In this way, the cells capture efficiency calculated may be more accuracy. The number of cells were in the range of 200 to 1200.
- The dispersion and electrospinning protocol for PLGA is missing in the Materials Section. It is suggested that the authors add this piece of information to provide additional information on the synthesis of the PLGA nanofibrous mesh over the micropillars.
Response: Thanks very much for the reviewer's suggestion. According to the reviewer’s suggestion, we had completed about the dispersion and electrospinning protocol of PLGA in the experiments and methods section.
Reviewer 4 Report
Comments from Reviewer
Title: Three dimensional PLGA nanofibers-based microchip for high efficiency cancer cells capture
The current form's presentation of methods and scientific results is satisfactory for publication in the Materials journal. The minor and significant drawbacks to be addressed can be specified as follows:
1. Line 20. Keywords?
2. Line 44. PLGA ---> PLGA (poly (lactic-co-glycolic acid))
3. Line 76. PDMS ---> PDMS (poly(dimethylsiloxane))
4. (i) Fig. 1(a) silicon ---> Silicon (ii) Fig. 1(d) silicon glass ---> Silicon Glass
5. Fig. 3, SEM. Please provide information/details about the instrument.
6. Line 234. (c) bright field ---> (c) Bright field
7. Fig. 5. What do the red arrows mean? Please complete the description of this figure..
8. Lines 252 and 253, “when the electrospinning time increased to 6 minutes, the capture efficiency was not significant increased.” Is a decrease of 10% a slight decrease? Were the measurements repeated? What do the error and error bars look like?
9. Fig. 6 (c) micropillars ---> Micropillars
10. References. Literature should also be standardized: the size of letters in the titles of journals, initials of names, the size of letters in the titles of articles. See titles of [1] and [2]
Sincerely
The reviewer.
Author Response
- Line 20. Keywords?
Response: Thanks very much for the reviewer's suggestion. We had added the key words in the revised manuscript.
Line 44. PLGA ---> PLGA (poly (lactic-co-glycolic acid))
Response: Thanks very much for the reviewer's suggestion. According to the reviewer’ suggestion, we had changed PLGA spelling rule in a normalized way, where PLGA first appeared in the manuscript.
Line 76. PDMS ---> PDMS (poly(dimethylsiloxane))
Response: Thanks very much for the reviewer's suggestion. According to the reviewer’ suggestion, we had changed PDMS spelling rule in a normalized way, where PDMS first appeared in the manuscript.
(i) Fig. 1(a) silicon ---> Silicon (ii) Fig. 1(d) silicon glass ---> Silicon Glass
Response: Thanks very much for the reviewer's suggestion. We had changed the word spelling in the revised manuscript according to the reviewer’s suggestion.
Fig. 3, SEM. Please provide information/details about the instrument.
Response: Thanks very much for the reviewer's suggestion. We had added experimental details about SEM characterization in the materials and methods section.
Line 234. (c) bright field ---> (c) Bright field
Response: Thanks very much for the reviewer's suggestion. We had changed the word spelling in the revised manuscript according to the reviewer’s suggestion.
Fig. 5. What do the red arrows mean? Please complete the description of this figure.
Response: Thanks very much for the reviewer's suggestion. We had added the red arrows mean in the revised manuscript.
Lines 252 and 253, “when the electrospinning time increased to 6 minutes, the capture efficiency was not significant increased.” Is a decrease of 10% a slight decrease? Were the measurements repeated? What do the error and error bars look like?
Response: Thanks very much for the reviewer's suggestion. As shown in fig.6 (a), when the electrospinning time increased to 6 minutes, the capture efficiency was not increased. There was a slight decrease, which may be mainly due to the large increased nanofibers. When the substrates were modified at the same condition, the density of antibodies modified on the nanofibers may be decreased, which may induce a decreased capture efficiency. This experiment was repeated. The cell capture efficiency was the average level. We had added the error bar in Fig.6 in the revised manuscript.
Fig. 6 (c) micropillars ---> Micropillars
Response: Thanks very much for the reviewer's suggestion. We had changed the word spelling in the revised manuscript according to the reviewer’s suggestion.
References. Literature should also be standardized: the size of letters in the titles of journals, initials of names, the size of letters in the titles of articles. See titles of [1] and [2]
Response: Thanks very much for the reviewer's suggestion. We had revised the references according to the journal standard.